# Uncovering women's healthcare access challenges in low- and middle-income countries using mixed effects modelling approach: Insights for achieving the Sustainable Development Goals

**Bewuketu Terefe**[1]*, **Belayneh Shetie Workneh**[2], **Gebreeyesus Abera Zeleke**[3], **Enyew Getaneh Mekonen**[3], **Alebachew Ferede Zegeye**[4], **Agazhe Aemro**[4], **Berhan Tekeba**[5], **Tadesse Tarik Tamir**[5], **Mulugeta Wassie**[4], **Mohammed Seid Ali**[5]

1 Department of Community Health Nursing, School of Nursing, College of Medicine and Health Sciences, University of Gondar, Gondar, Ethiopia, 2 Department of Emergency and Critical Care Nursing, School of Nursing, College of Medicine and Health Sciences, University of Gondar, Gondar, Ethiopia, 3 Department of Surgical Nursing, School of Nursing, College of Medicine and Health Sciences, University of Gondar, Gondar, Ethiopia, 4 Department of Medical Nursing, School of Nursing, College of Medicine and Health Sciences, University of Gondar, Gondar, Ethiopia, 5 Department of Pediatrics and Child Health Nursing, School of Nursing, College of Medicine and Health Sciences, University of Gondar, Gondar, Ethiopia

* woldeabwomariam@gmail.com

## Abstract

### Background

Access to healthcare services for women in low- and middle-income countries (LMICs) is crucial for maternal and child health and achieving the Sustainable Development Goals (SDGs). However, women in LMICs face barriers to accessing healthcare, leading to poor health outcomes. This study used Demographic and Health Survey (DHS) data from 61 LMICs between 2010–2023 to identify women's healthcare access challenges.

### Methods

This study used data from the DHS conducted in 61 LMICs to identify women's healthcare access challenges from 2010 to 2023. A weighted sample of 1,722,473 women was included in the study using R-4.4.0 version software. A mixed-effects modeling approach was used to analyze access to healthcare, considering individual-level factors and contextual factors. The mixed-effects model takes into account clustering within countries and allows for the examination of fixed and random effects that influence women's healthcare access across LMICs. For the multivariable analysis, variables with a p-value ≤0.2 in the bivariate analysis were considered. The Adjusted Odds Ratio (AOR) with a 95% Confidence Interval (CI) and a P value < 0.05 was reported to indicate statistical significance and the degree of association in the final model.

**Data Availability Statement:** Third party data was obtained for this study from DHS Program. Data

may be requested from DHS Program after creating an account and submitting a concept note. More access information can be found on the DHS Program website (https://dhsprogram.com/data/Access-Instructions.cfm). The authors confirm that interested researchers would be able to access these data in the same manner as the authors. The authors also confirm that they had no special access privileges that others would not have.

**Funding:** The author(s) received no specific funding for this work.

**Competing interests:** The authors have declared that no competing interests exist.

## Results

The pooled prevalence of the healthcare access problem was found to be 66.06 (95% CI: 61.86, 70.00) with highly heterogeneity across countries and regions. Women aged 25–34 years, and 35–49 years, had primary education, and secondary or higher education, married women, poorer, middle, richer, and richest wealth indices, had mass media exposure, first birth at age $\geq$20 years, birth interval of 24–36, 37–59 and >59 months as compared to < 24 months birth interval, had health insurance, delivered at a health facility, had at least one ANC visit, being from lower-middle-income countries, upper-middle-income countries, regions like West Africa, South Asia, and East Asia/Pacific compared to women living in East Africa, low literacy rates, medium literacy rates, and high literacy rates as compared to very low literacy rate were associated with lower odds of having problems accessing healthcare respectively. On the other hand, divorced/widowed women, having 1–2, and more than two under five, living in households with 6–10 family members and >10 members, female household heads, living in rural areas, women living in South/Central Africa, Middle East/North Africa, Europe/Central Asia, and living in Latin America/Caribbean were associated with higher odds of having problems accessing healthcare respectively.

## Conclusions

Approximately two-thirds of women face healthcare access problems. Sociodemographic factors such as age, education, marital status, wealth, media exposure, and health insurance are associated with lower odds of experiencing healthcare access issues. On the other hand, factors such as divorce/widowhood, the number of young children, household size, female household heads, rural residence, and region have been linked to higher odds of facing healthcare access challenges. To address these disparities, policies, and interventions should focus on vulnerable populations by improving access to health insurance, increasing educational attainment, and providing support for single mothers and large households. Additionally, tailored regional approaches may be necessary to overcome barriers to healthcare access.

## Introduction

In the pursuit of global health equity, ensuring access to healthcare services for women is imperative. Women's healthcare access is not only a fundamental human right but also a crucial determinant of their overall well-being and that of their families and communities [1,2]. However, despite significant progress in healthcare infrastructure and policy reforms, numerous challenges persist, particularly in low- and middle-income countries (LMICs) [3]. Globally, in 2017, half of the population lacked basic health services, per the World Bank and World Health Organization (WHO) [4,5]. In Africa, 11 million people fall into poverty due to healthcare expenses [4]. Accessibility and availability of essentials vary greatly in Sub-Saharan Africa (SSA) and Southern Asia [5,6]. Understanding and addressing these challenges are essential steps towards achieving the Sustainable Development Goals (SDGs) set forth by the United Nations, particularly SDG 3, which aims to ensure healthy lives and promote well-being for all at all ages [6,7].

Despite the growing body of literature on women's health and healthcare access, there are still notable gaps in our understanding of the challenges faced by women in LMICs. Firstly,

existing research often focuses on specific regions or individual countries, providing limited insights into the broader landscape of women's healthcare access across LMICs [8–10]. As a result, there is a lack of comprehensive analyses that encompass a wide range of countries with diverse socio-economic and cultural contexts.

Secondly, while some studies have examined the barriers to healthcare access for women in LMICs, there is a dearth of research utilizing advanced statistical techniques to account for the complex interplay of factors influencing access. Mixed effects modeling offers a powerful approach to address this gap by incorporating both individual-level and contextual factors into the analysis, thereby providing a more nuanced understanding of the determinants of healthcare access for women.

Furthermore, previous studies did not incorporate important factors like the income level of the country, literacy rate of the country, the sex of the household head, family member numbers, and under-five children in the household, contraception utilization, and women's decision-making ability regarding their healthcare needs [11–13]. These factors are crucial in understanding the multifaceted nature of healthcare access challenges faced by women in LMICs.

Furthermore, while the importance of achieving the SDGs, particularly SDG 3, is widely recognized, there is a need for more research that directly examines the implications of healthcare access challenges for the achievement of these goals [12,13]. Several regional and country studies such as those from sub SSA [12], Ethiopia [14], Rwanda [15], and resource limited settings [16] have revealed that several factors are likely to obstruct women's access to healthcare and an especially low-income level. Specifically, these barriers include transportation, geographical location, system organizational barriers, the general availability of services, health information, waiting times, and health infrastructure [12,16]. By bridging this gap, this study aims to provide actionable insights that can inform policy decisions and interventions aimed at improving women's healthcare access and advancing progress towards the SDGs in LMICs.

This research aims to explore the healthcare access challenges faced by women in 61 LMICs. By employing a mixed effects modeling approach, this study seeks to explore the multifaceted factors influencing women's access to healthcare services in diverse socio-cultural and economic contexts. By uncovering these challenges and their underlying determinants, the research aims to provide valuable insights that can inform evidence-based policy interventions and strategies to improve women's healthcare access in LMICs. Ultimately, the findings of this study aim to contribute to the global efforts towards achieving the SDGs, particularly in the realm of women's health.

## Methods

### Study setting and period

The study was conducted across 61 LMICs, spanning multiple geographic regions from 2010 to 2023 using the nationwide community-based cross-sectional studies. These countries were categorized into eight distinct subregions: East Africa, West Africa, South/Central Africa, Middle East, North Africa, South Asia, Europe, Central Asia, Latin America, Caribbean, and East Asia/Pacific. The selection of these 61 countries and their categorization into the eight subregions was informed by established classifications from authoritative sources such as the WHO, Demographic and Health Surveys (DHS), and other major international health databases and indicators. This broad geographic coverage allowed the researchers to examine the determinants of women's access to healthcare services across a diverse array of LMICs contexts, providing valuable insights that can inform policies and programs to improve healthcare equity in these regions.

## Source and study population

The source population for this study comprises women aged 15–49 years residing in LMICs across the world. The study population, on the other hand, consists of a subset of women aged 15–49 years who participated in the most recent DHS survey conducted in these countries between 2010 and 2023.

## Sample size determination and methods

Every five years, DHSs are compiled in LMICs. These surveys used validated questionnaires and a standardized methodology for sampling, data collection, and coding. We based our surveys on the most recent census frames in all listed countries. The DHS samples were divided into urban and rural areas within each administrative region. In developing countries, the statistics are often incomplete or outdated. To ensure the precise implementation and control of fieldwork, we followed a simple sampling design as our policy at Macro. For the majority of the DHSs, we employed a two-stage cluster sampling approach. In the first stage, we selected primary sampling units (PSUs) based on their size. In the second stage, we selected a fixed number of households from a list obtained during an update operation from the chosen PSUs.

A PSU is an Enumeration Area (EA) that is a specific area or part of an area. EAs are created based on the most recent population census and usually include multiple households. A comprehensive list of EAs provides information on their geographical location, rural-urban characteristics, population, and households. The boundaries of the EAs are clearly defined in the available cartographic materials. However, since a population census is conducted every five years, EA information, such as the number of households, may be outdated and needs to be updated. The updating process involves listing all households in selected EAs and recording key information, such as the household head's name, street address, and residence type. This creates a comprehensive list of households in selected EAs, which serves as the sampling frame for second-stage household selection [17]. After the households have been listed, a method called equal probability systematic sampling is employed to choose a specific number of households from within the designated cluster [18].

## Data source

The data used in this study were obtained from the most recent DHS dataset for 61 LMICs after a formal online dataset request using https://www.dhsprogram.com/data/dataset_admin address. The DHS program ensures the use of standardized data collection methods across all countries [18]. To ensure a substantial and inclusive sample size that considers multiple factors, the DHS office conducts surveys on a global level. These surveys are specifically designed to collect comparable data and utilize large sample sizes. They are conducted at the population level, with the aim of providing nationally representative information for each country [18]. In each country's survey, distinct datasets were collected for men, women, infants, births, and households. For this specific analysis, we utilized the women's datasets (IR files). The study primarily focused on women aged 15–49. For the analysis, we collected and combined the women's data from the most recent DHS conducted in 61 countries. In total, a weighted sample of 1,718,793 women was utilized, representing a diverse range of participants from the regions. This information offers an overview of the sample size and participant distribution within the study, facilitating a comprehensive investigation of women-related factors in LMICs using the pooled DHS data.

## Variables of the study

**Outcome variable.** The main outcome variable of interest in this study was women's accessibility to healthcare. The researchers in this study incorporated important factors like

travel time and transportation costs when examining women's access to healthcare facilities. The researchers utilized data from the DHS, which asked women a series of questions about potential barriers they might face when trying to access healthcare. From these DHS survey questions; the researchers generated a composite outcome variable to measure healthcare accessibility. The specific DHS questions included whether getting the money needed for treatment, the distance to a healthcare facility, having to take transport, and not wanting to go alone would be a big problem or not a big problem. For each question, the response options were "big problem" or "not a big problem". If a woman indicated that at least one of these factors would be a big problem for her, then her access to healthcare was coded as 1, meaning it was considered a big problem. If she did not identify any of these as a big problem, then her access to healthcare was coded as 0, meaning it was not a big problem [19]. By creating this composite measure of healthcare accessibility based on multiple potential barriers, the researchers were able to take a more comprehensive approach compared to studies that may have only focused on one or two accessibility factors. This allowed them to gain a fuller understanding of the various challenges women face when trying to access needed healthcare services.

**Independent variables.**   The researchers took a thorough and comprehensive approach to selecting the explanatory variables for this study. We reviewed various literature, guidelines, and scientific facts before determining which variables to incorporate [3,9,12–14]. After this review process, we retrieved relevant variables from the DHS dataset. The variables considered spanned multiple levels, including the individual, community, and regional levels. At the individual level, the variables included factors such as the woman's age group, educational status, marital status, occupation, media exposure, wealth status, birth order, birth interval, age at first birth, antenatal care utilization, place of delivery, number of under five children in the household, family members, contraception use, sex of the household head, health insurance coverage, and the primary decision-maker for the woman's healthcare needs. The community-level variables examined included the woman's place of residence, the sub-region within the country, the overall income level of the country, and the literacy rate in the woman's country. By taking this multilevel approach and thoroughly reviewing the potential explanatory factors, the researchers were able to develop a robust and comprehensive analysis of the various individual, community, and regional determinants of women's access to healthcare services.

## Operational definitions

**Country literacy rate.**   Evidence from the World Bank and World Population Review indicates that developed nations have an average literacy rate of more than 90%, while least developed nations have an average literacy rate of only 65% [20,21]. Using this as a baseline, the researchers divided the countries' literacy rates into the following categories: High Literacy: Countries with a literacy rate greater than 90%, Medium Literacy: Countries with a literacy rate between 75–90%, Low Literacy: Countries with a literacy rate between 65–75%, and Very Low Literacy: Countries with a literacy rate less than 65%.

**Subregions.**   Based on the classification of the WHO, DHS, and other international health indicators [19,22], the countries were grouped according to the following classification: East Africa, West Africa, South/central Africa, middle east/north Africa, South Asia, Europe/Central Asia, Latin America/Caribbean, and East Asia/ Pacific.

**Country's income level.**   The variable 'country income' was calculated using the World Bank classification, which divides countries into low income, lower middle income, and upper middle-income categories [23].

**Mass media exposure.**   This variable was generated from three possible media sources: watching television, listening to radio, and reading magazines/books. Women's who were

exposed to at least one of these media sources were recoded as having mass media exposure. Those who were not exposed to any of the three media sources were recoded as having no mass media exposure.

## Data management and statistical analyses processes

The data were downloaded using STATA version17, and then it was cleaned and analyzed using R-4.4.0 version software and Microsoft Excel version 19. Descriptive data, including frequencies and percentages of various variables, are presented using text, tables, and graphs. Multilevel logistic regression models were used to examine the associations between each independent variable and outcome variable. Variables with p-values $\leq 0.2$ were considered for analysis in the univariate analysis. In the final model, the adjusted odds ratio was used to assess the relationship between the dependent and independent variables, with variables considered statistically significant if their p-value was less than 0.05. In the DHS data, each woman was assigned to a specific cluster, and women within the same cluster demonstrated greater similarity than those in different clusters. This violates the assumptions of observational independence and equal variance across clusters in a standard regression model. Therefore, a more complex multilevel random intercept logistic regression model was employed to account for the between-cluster effects. This multilevel random intercept logistic regression model was developed to investigate the association between individual-level and community-level factors and the likelihood of a woman not having access to healthcare or having it.

Four models were generated for analysis. The first model, also referred to as an empty or null model, was fitted without explanatory variables. This approach aimed to minimize the differences between communities and provide insights into community variance. Understanding the null model is essential for understanding the influence of sociodemographic factors on risky sexual behaviors in women. Additionally, this model served as a reference for adopting a multilevel statistical framework and comparing it with a conventional logistic regression. Various statistical measures, including Proportional Change of Variance (PCV), Log-Likelihood Ratio test (LLR), Median Odds Ratio (MOR), Intraclass Correlation Coefficient (ICC), and AIC, were used to analyze the null model. The second model incorporated only individual-level characteristics, whereas the third model included only neighborhood-level features. In contrast, the final (fourth) model included both individual- and community-level components. The model with the lowest deviation was selected for reporting and interpreting the results when comparing the models using model deviance.

Deviance (2log likelihood) was used to compare stacked models. The variance between clusters was calculated using log-likelihood and intraclass correlation coefficient (ICC). The ICC indicates the level of variation among women. To determine the individual- and community-level factors influencing the outcome variable among women, multilevel logistic regression analysis was conducted. Model four was found to have the highest value and best fit when the models were compared using the likelihood test. The measure of variation was evaluated using the median odds ratio (MOR), which is the median value of the odds ratio between the area at the lowest risk and the highest risk when two clusters are randomly selected. MOR = e0.95$\sqrt{}$VA or, MOR = exp. [$\sqrt{}$ (2 × VA) × 0.6745], where; VA is the area level variance [24,25]. The Proportional Change in Variance (PCV) reveals the variation in accessing healthcare among women explained by factors. The PCV is calculated as $= \frac{Vnull-VA}{Vnull} *100$. Where: Vnull is the initial model's variance and VA is the model's variance with additional terms. Also, the ICC, a measurement of the variation in accessing healthcare between clusters, is computed as; ICC = VA÷VA+3.29 *100%, where; VA = area/cluster level variance [24,25].

### Ethical considerations and data set access

The study was conducted after obtaining a permission letter from www.dhsprogram.com on an online request to access LMICs DHS data after reviewing the submitted brief descriptions of the survey to the DHS program. The datasets were treated with the utmost confidence. This study was done based on secondary data from LMICs DHS. Issues related to informed consent, confidentiality, anonymity, and privacy of the study participants are already done ethically by the DHS office. We did not manipulate and apply the microdata other than in this study. There was no patient or public involvement in this study.

## Results

### Sociodemographic characteristics of participants

The majority of women are aged 15–24 years 596,347 (34.70%). Most women had a secondary or higher level of education 983,971 (57.25%), and over half were not working 896,795 (53.43%). The largest wealth index group was the richest at 371,534 (21.62%). The majority of women had exposure to mass media 1,049,426 (61.06%) and did not have any children under 5 867,546 (50.47%). Most women had their first birth at age 20 or older 1,145,237 (66.63%), and the most common birth interval is over 59 months 957,826 (55.73%). The majority of women did not use modern contraceptives 1,014,897 (59.05%), and healthcare decisions were primarily made by the woman herself (1,543,413 89.80%). Most households did not have health insurance 1,290,000 (77.47%) and have 1–5 family members 957,990 (55.74%). Over 60% of women were currently married 1,073,159 (62.44%), and the vast majority delivered their babies at a health facility 1,585,262 (92.23%). Nearly all women received antenatal care 1,661,083 (96.64%), and most households were headed by a male 1,358,580 (79.04%). Regarding the regional and country-level characteristics, the majority of women reside in rural areas 1,041,011 (60.57%). In terms of country income levels, the largest group was from lower-middle-income countries 1,219,102 (70.93%). Looking at the subregions, the largest representation was from South Asia 821,496 (47.79%). About the country literacy rates, the majority of women were from countries with low literacy rates (904,787 (52.64%) (Table 1).

### Pooled prevalence of healthcare access problem among women in LMICs

The forest plot from a meta-analysis showed women's access to healthcare across 61 LMICs. This forest plot examined the effect size (ES) which represents for healthcare access problems among women in each country. The effect size is a measure of the magnitude of the issue, with a higher number indicating more significant access problems. The plot also showed the 95% confidence interval (CI) for each country's effect size, which indicates how precise the estimate is.

The countries were grouped into different geographic regions, like South Asia, Latin America/Caribbean, and Europe/Central Asia. For each region, there is a summary effect size and confidence interval calculated using a random effects model. Moreover, the regional summaries, revealed that the region with the highest effect size was Latin America/Caribbean, with an ES of 76.77 (95% CI: 74.29, 81.08). This suggested very significant barriers to healthcare access for women in that part of the world. In contrast, the region with the lowest effect size was Europe/Central Asia, with an ES of 52.63 (95% CI: 41.37, 61.38). Across all regions, the overall random effects model showed an effect size of 66.06 (95% CI: 61.86, 70.00). finally, the forest plot also revealed a high degree of heterogeneity, as indicated by the $I^2$ statistic being 100% for all regions and the overall model. This means the effect sizes varied widely between countries (Fig 1).

**Table 1. Socio-demographic, maternal, and community level related characteristics of women's accessing healthcare among women in LMICs (weighted n = 1,718,793).**

| Variables | Categories | Frequency | Percentage |
|---|---|---|---|
| Women age in years | 15–24 | 596,347 | 34.70 |
| | 25–34 | 529,584 | 30.81 |
| | 35–49 | 592,862 | 34.49 |
| Women's educational status | No education | 380,215 | 22.12 |
| | Primary | 354,607 | 20.63 |
| | Secondary and higher | 983,971 | 57.25 |
| Women's occupation | Not working | 896,795 | 53.43 |
| | Working | 821,998 | 46.57 |
| Wealth index | Poorest | 308,046 | 17.92 |
| | Poorer | 332,085 | 19.32 |
| | Middle | 345,590 | 20.11 |
| | Richer | 361,539 | 21.03 |
| | Richest | 371,534 | 21.62 |
| Mass media exposure | No | 669,368 | 38.94 |
| | Yes | 1049426 | 61.06 |
| Number of under five children | No | 867,546 | 50.47 |
| | 1–2 | 731,900 | 42.58 |
| | >2 | 119,347 | 6.94 |
| Age at first birth | <20 | 573,556 | 33.37 |
| | ≥20 | 1145237 | 66.63 |
| Preceding birth intervals in months | <24 | 211,207 | 12.29 |
| | 24–36 | 290,630 | 16.91 |
| | 37–59 | 259,130 | 15.08 |
| | >59 | 957,826 | 55.73 |
| Modern contraceptive | No | 1014897 | 59.05 |
| | Yes | 703,896 | 40.95 |
| Who decided on women healthcare needs | Woman herself | 154,3413 | 89.80 |
| | Not the woman | 175,381 | 10.20 |
| Health insurance | No | 1,290,000 | 77.47 |
| | Yes | 428,793 | 22.53 |
| Family member in number | 1–5 | 957,990 | 55.74 |
| | 6–10 | 639,901 | 37.23 |
| | >10 | 120,903 | 7.03 |
| Marital status | Never married | 425,984 | 24.78 |
| | Married | 1073159 | 62.44 |
| | Divorced/widowed | 219,650 | 12.78 |
| Place of delivery | Home | 133,532 | 7.77 |
| | Health facility | 1585262 | 92.23 |
| ANC visit | No | 57,711 | 3.36 |
| | Yes | 166,1083 | 96.64 |
| Sex of the household head | Male | 135,8580 | 79.04 |
| | Female | 360,213 | 20.96 |
| Residence | Urban | 677,783 | 39.43 |
| | Rural | 1041011 | 60.57 |

*(Continued)*

**Table 1.** (Continued)

| Variables | Categories | Frequency | Percentage |
|---|---|---|---|
| Country's income level | Low | 235,116 | 13.68 |
| | Lower middle | 1219102 | 70.93 |
| | Upper middle | 264,575 | 15.39 |
| Subregion | East Africa | 178,730 | 10.40 |
| | West Africa | 191,632 | 11.15 |
| | South/central Africa | 105,123 | 6.12 |
| | Middle east/north Africa | 83,647 | 4.87 |
| | South Asia | 821,496 | 47.79 |
| | Europe/Central Asia | 66,257 | 3.85 |
| | Latin-American/Caribbean | 147,160 | 8.56 |
| | East Asia/Pacific | 124,749 | 7.26 |
| Country's literacy rate | Very low | 235,522 | 13.70 |
| | Low | 904,787 | 52.64 |
| | Medium | 357,862 | 20.82 |
| | High | 220,622 | 12.84 |

## Random parameters analysis

The null model, which serves as the baseline, shows a community-level variance of 2.29. This indicated a significant amount of variation in women's healthcare access at the community level. The ICC for the null model was 40.93%, meaning that 40.93% of the total variance in healthcare access was attributable to differences between communities. Moving to the subsequent models, Model I showed a reduction in the community-level variance to 1.89, a 17.47% decreased from the null model. This suggested that the inclusion of individual-level variables in this model was able to explain some of the between-community variation in healthcare access. The ICC for Model I was 36.45%, indicating that 36.45% of the total variance was still explained by community-level factors. In Model II, the community-level variance remains the same as in Model I at 1.89, and the ICC is 36.55%. This implies that the addition of household-level variables in this model did not significantly improve the explanation of between-community variation compared to Model I. Finally, Model III revealed the lowest community-level variance of 1.69, a 26.20% reduction from the null model. The ICC for this model was 33.97%, suggesting that 33.97% of the total variance in healthcare access was still attributable to community-level factors. This model, with the inclusion of both individual- and household-level variables, appeared to be the best-fitting model, as indicated by the lower values for the Log-Likelihood Ratio (LLR), Deviance Information Criterion (DIC), and Akaike Information Criterion (AIC) compared to the other models. Overall, the results indicated that both individual- and community-level factors play a significant role in determining women's healthcare access in LMICs, with the community-level factors accounting for a substantial portion of the total variance (Table 2).

## Factors associated with healthcare access problems among women in LMICs

Several individual and community-level factors associated with women's access to healthcare challenges in LMICs. According to the AOR model with a 95% CI, women aged 25–34 years had 0.92 (95% CI: 0.91–0.93) times lower odds of having problems accessing healthcare compared to the reference group of women aged 15–24 years. Similarly, women aged 35–49 years

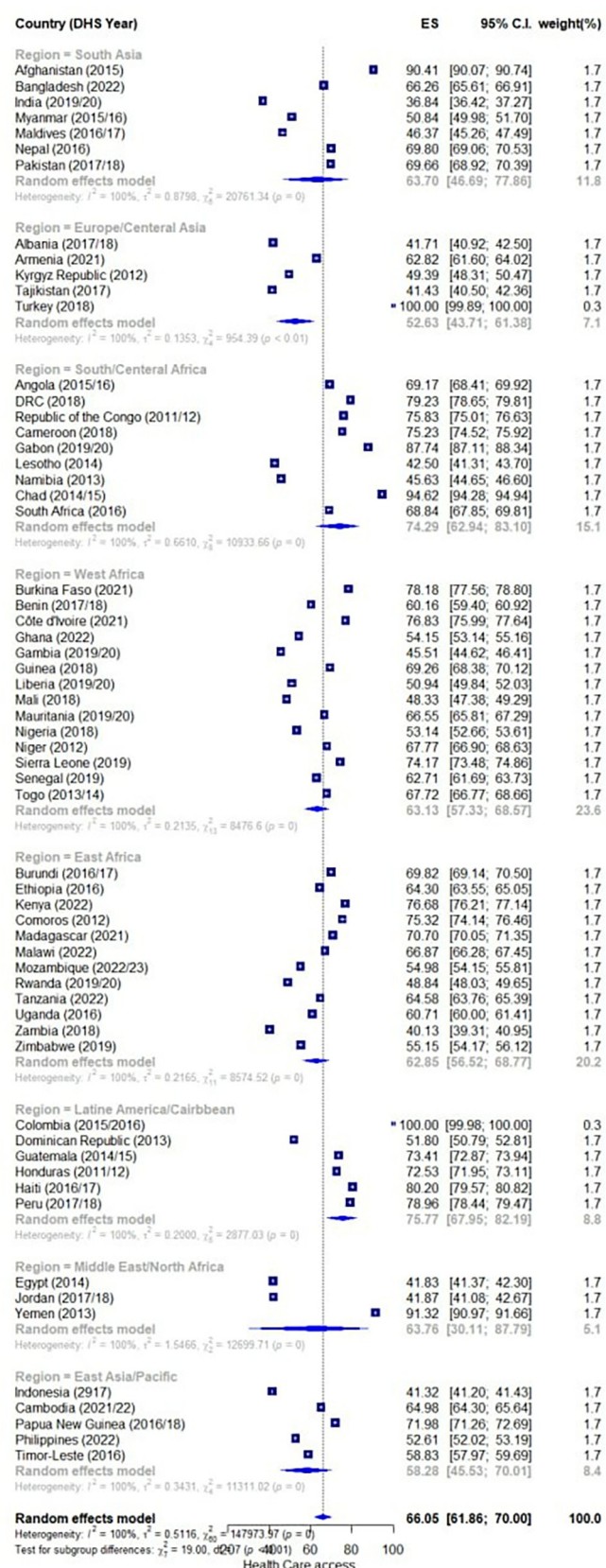

**Fig 1. Pooled prevalence of healthcare access problem among women in LMICs.**

**Table 2. Random parameters and model comparison among women's accessing healthcare in LMICs.**

| Random parameters and model comparison | Null model | Model I | Model II | Model III |
|---|---|---|---|---|
| Community level variance | 2.29 | 1.89 | 1.89 | 1.69 |
| ICC | 40.93 | 36.45 | 36.55 | 33.97 |
| MOR | 4.20 | 3.69 | 3.69 | 3.44 |
| PCV | Reference | 17.47 | 17.47 | 26.20 |
| LLR | -1060386 | -1015174 | -1020982 | -990902 |
| DIC | 2120772 | 2030348 | 2041964 | 1981804 |
| AIC | 2120774 | 2030399 | 2041993 | 1981880 |

had 0.89 (95% CI: 0.88–0.91) times lower odds of having problems accessing healthcare compared to the reference group. Women with primary education had 0.82 (95% CI: 0.81–0.83) times lower odds of having problems accessing healthcare compared to the reference group of women without education. Additionally, women with secondary or higher education had 0.65 (95% CI: 0.65–0.66) times lower odds of having problems accessing healthcare compared to the reference group. Married women had 0.87 (95% CI: 0.86–0.88) times lower odds of having problems accessing healthcare compared to never-married women. In contrast, divorced/widowed women had 1.09 (95% CI: 1.07–1.11) times higher odds of having problems accessing healthcare compared to never-married women. The model demonstrated that as wealth increases from the poorest to the richest quintile, the odds of having problems accessing healthcare decreased. Specifically, the richest women had 0.31 (95% CI: 0.31–0.32) times lower odds of having problems accessing healthcare compared to the poorest women. Women with 1–2 children under five had 1.02 (95% CI: 1.01–1.02) times higher odds of having problems accessing healthcare compared to women with no children under five. Women with two or more children under five had 1.04 (95% CI: 1.02–1.06) times higher odds of having problems accessing healthcare compared to their counterparts. Women living in households with 6–10 family members had 1.03 (95% CI: 1.02–1.04) times higher odds of having problems accessing healthcare compared to women living in households with 1–5 family members. Similarly, women living in households with more than 10 family members had 1.05 (95% CI: 1.04–1.07) times higher odds of having problems accessing healthcare compared to the reference group. Women with mass media exposure had 0.86 (95% CI: 0.85–0.87) times lower odds of having problems accessing healthcare compared to women without mass media exposure. Women who had their first birth at age 20 or older had 0.94 (95% CI: 0.93–0.95) times lower odds of having problems accessing healthcare compared to women who had their first birth before age 20. Women with a birth interval of 24–36 months had 0.98 (95% CI: 0.96–0.99) times lower odds of having problems accessing healthcare compared to women with a birth interval of less than 24 months. Women with a birth interval of 37–59 months had 0.97 (95% CI: 0.96–0.98) times lower odds of having problems accessing healthcare compared to women with a shorter birth interval. Women with a birth interval of 59 months or more had 0.93 (95% CI: 0.92–0.95) times lower odds of having problems accessing healthcare compared to the reference group. Women with health insurance had 0.65 (95% CI:0.64,0.66) times lower odds of having problems accessing healthcare compared to women without health insurance. Women who delivered at a health facility had 0.85 (95% CI: 0.84–0.87) times lower odds of having problems accessing healthcare compared to women who delivered at home. Women who had at least one ANC visit have 0.75 (95% CI: 0.73–0.77) times lower odds of having problems accessing healthcare compared to women who did not have any ANC visits. Women living in households with a female head had 1.02 (95% CI: 1.01–1.03) times higher odds of having problems accessing healthcare compared to women living in households with a male head.

Regrading community level related factors, women living in rural areas had 1.14 (95% CI: 1.12–1.15) times higher odds of having problems accessing healthcare compared to women living in urban areas. Women living in lower-middle-income countries had 0.78 (95% CI: 0.76–0.79) times lower odds of having problems accessing healthcare compared to women living in low-income countries. Women living in upper-middle-income countries had 0.75 (95% CI: 0.73–0.77) times lower odds of having problems accessing healthcare compared to the reference group. Women living in West Africa had 0.76 (95% CI: 0.74–0.77) times lower odds of having problems accessing healthcare compared to women living in East Africa. Women living in South/Central Africa had 1.68 (95% CI: 1.65–1.72) times higher odds of having problems accessing healthcare compared to the reference group. Women living in the Middle East/ North Africa had 2.22 (95% CI: 2.17–2.29) times higher odds of having problems accessing healthcare compared to the reference group. Women living in South Asia had 0.98 (95% CI: 0.96–0.99) times lower odds of having problems accessing healthcare compared to the reference group. Women living in Europe/Central Asia had 1.57 (95% CI: 1.54–1.61) times higher odds of having problems accessing healthcare compared to the reference group. Women living in Latin America/Caribbean had 2.58 (95% CI: 2.53–2.63) times higher odds of having problems accessing healthcare compared to the reference group. Women living in East Asia/Pacific had 0.97 (95% CI: 0.95–0.98) times lower odds of having problems accessing healthcare compared to the reference group. Finally, women living in countries with low literacy rates had 0.84 (95% CI: 0.83–0.85) times lower odds of having problems accessing healthcare compared to women living in areas with very low literacy rates. Women living in areas with medium literacy rates had 0.79 (95% CI: 0.78–0.81) times lower odds of having problems accessing healthcare compared to the reference group. Women living in areas with high literacy rates had 0.38 (95% CI: 0.37–0.39) times lower odds of having problems accessing healthcare compared to the reference group (Table 3).

## Discussion

This study was conducted to explore the individual and contextual level factors of women's healthcare access challenges in LMICs to support the aim of the SDGs by showing the magnitude of the problem among these vulnerable segments of the population. In the final model, several individual, countries, and regional-level factors were found to be statistically significant, including age, education, marital status, wealth, family size, media exposure, birth history, and health insurance coverage, which play a significant role in determining women's access to healthcare in LMICs. Community-level factors include geographic region, income level, and literacy rate. In the random forest plot analysis, we found that the overall magnitude of healthcare access problem was found 66.06 (95% CI: 61.86, 70.00), which was more than two-thirds of women faced the challenge. In summary, the forest plot provided a comprehensive look at the troubling state of women's access to healthcare across a large number of developing countries. These results highlighted the need for targeted, context-specific interventions to improve access and reduce inequities. Policymakers and public health practitioners will likely find this meta-analysis valuable for identifying priority areas and designing effective strategies to support women's health.

Young women, particularly those aged 15–24, had higher odds of having problems accessing healthcare compared to older women due to several factors. Young women are more likely to face permission-related barriers, such as needing permission from parents or guardians to seek healthcare or treatment advice [26,27]. While they have lower odds of financial barriers compared to older women, young women are still more likely to face financial constraints due to limited resources [26–28]. Distance-related barriers, such as difficulty accessing healthcare

**Table 3. Individual and community-level factors associated with women's accessing healthcare among women in LMICs.**

| Accessing healthcare | Null model | Model I AOR, 95% CI | Model III AOR, 95% CI | Model III AOR, 95% CI |
|---|---|---|---|---|
| **Variables** | | | | |
| **Maternal age** | | | | |
| 15–24 | | 1 | | 1 |
| 25–34 | | 0.89(0.89,0.91) | | 0.92(0.91,0.93) * |
| 35–49 | | 0.87(0.86,0.89) | | 0.89(0.88,0.91) * |
| **Maternal education** | | 1 | | |
| Not educated | | 1 | | 1 |
| Primary | | 0.82(0.81,0.83) | | 0.82(0.81,0.83) * |
| Secondary & higher | | 0.64(0.63,0.65) | | 0.65(0.65,0.66) * |
| **Marital status** | | | | |
| Never married | | 1 | | 1 |
| Married | | 0.80(0.79,0.81) | | 0.87(0.86,0.88) * |
| Divorced/widowed | | 1.20(1.19,1.22) | | 1.09(1.07,1.11) * |
| **Wealth index** | | | | |
| Poorest | | 1 | | 1 |
| Poorer | | 0.77(0.76,0.78) | | 0.77(0.76,0.78) * |
| Middle | | 0.60(0.59,0.61) | | 0.61(0.60,0.62) * |
| Richer | | 0.46(0.46,0.47) | | 0.48(0.47,0.48) * |
| Richest | | 0.30(0.29,0.31) | | 0.31(0.31,0.32) * |
| **Number of under five** | | | | |
| No | | 1 | | 1 |
| 1–2 | | 1.01(1.01,1.02) | | 1.02(1.01, 1.02) * |
| >2 | | 1.03(1.02,1.06) | | 1.04(1.02,1.06) * |
| **Family members** | | | | |
| 1–5 | | 1 | | 1 |
| 6–10 | | 1.04(1.04,1.06) | | 1.03(1.02, 1.04) * |
| >10 | | 1.09(1.07,1.11) | | 1.05(1.04,1.07) * |
| **Mass media exposure** | | | | |
| No | | 1 | | 1 |
| Yes | | 0.93(0.92,0.94) | | 0.86(0.85,0.87) * |
| **Age at first birth** | | | | |
| <20 | | 1 | | 1 |
| ≥20 | | 0.91(0.90,0.92) | | 0.94(0.93,0.95) * |
| **Birth interval** | | | | |
| <24 | | 1 | | 1 |
| 24–36 | | 0.96(0.95,0.98) | | 0.98(0.96,0.99) * |
| 37–59 | | 0.94(0.93,0.96) | | 0.97(0.96,0.98) * |
| >59 | | 0.91(0.90,0.93) | | 0.93(0.92,0.95) * |
| **Health insurance** | | | | |
| No | | 1 | | 1 |
| Yes | | 0.66(0.64,0.67) | | 0.65(0.64,0.66) * |
| **Place delivery** | | | | |
| Home | | 1 | | 1 |
| Health facility | | 0.76(0.74,0.77) | | 0.85(0.84,0.87) * |
| **ANC visit** | | | | |
| No | | 1 | | 1 |

*(Continued)*

**Table 3.** (Continued)

| Accessing healthcare | Null model | Model I AOR, 95% CI | Model III AOR, 95% CI | Model III AOR, 95% CI |
|---|---|---|---|---|
| Yes | | 0.67(0.66,0.69) | | 0.75(0.73,0.77) * |
| **Modern contraceptive** | | | | |
| No | | 1 | | 1 |
| Yes | | 1.01(1.01,1.03) | | 0.95(0.97, 1.01) |
| **Household head** | | | | |
| Male | | 1 | | 1 |
| Female | | 1.02(1.01,1.03) | | 1.02(1.01,1.03) * |
| **Residence** | | | | |
| Urban | | | 1 | 1 |
| Rural | | | 2.01(1.99,2.3) | 1.14(1.12,1.15) * |
| **Income level** | | | | |
| Low | | | 1 | 1 |
| Lower middle | | | 0.77(0.76,0.78) | 0.78(0.76,0.79) * |
| Upper middle | | | 0.74(0.73,0.76) | 0.75(0.73,0.77) * |
| **Subregions** | | | | |
| East Africa | | | 1 | 1 |
| West Africa | | | 0.88(0.86,0.89) | 0.76(0.74,0.77) * |
| South/central Africa | | | 2.03(1.99,2.07) | 1.68(1.65,1.72) * |
| Middle east/north Africa | | | 2.20(2.15,2.26) | 2.22(2.17,2.29) * |
| South Asia | | | 0.99(0.97,1.01) | 0.98(0.96,0.99) * |
| Europe/Central Asia | | | 1.41(1.38,1.45) | 1.57(1.54,1.61) * |
| Latin-American/Caribbean | | | 2.92(2.86,2.98) | 2.58(2.53,2.63) * |
| East Asia/Pacific | | | 0.93(0.91,0.95) | 0.97(0.95,0.98) * |
| **Literacy rate** | | | | |
| Very low | | | 1 | 1 |
| Low | | | 0.75(0.74,0.76) | 0.84(0.83,0.85) * |
| Medium | | | 0.74(0.73,0.75) | 0.79(0.78,0.81) * |
| High | | | 0.37(0.37,0.38) | 0.38(0.37,0.39) * |

Where * = statistically significant variables in the final model.

facilities, can also be more challenging for young women, especially those living in rural areas [27,28]. Social and cultural factors, including not wanting to present to healthcare alone or feeling uncomfortable discussing certain health issues, can pose additional barriers that are more prevalent among younger women. However, young women with higher education and employment opportunities are more likely to have lower odds of healthcare barriers, as education and employment can provide greater financial stability, social support, and access to healthcare resources.

The findings also revealed that the educational attainment of the woman was a significant determinant of healthcare access. Women with higher levels of education are more likely to access healthcare services compared to those with lower levels of education. This aligns with the findings from previous studies conducted in various countries, as well as the WHO global health survey and studies in South Asia and SSA [12,29–31]. The possible explanation for this could be that education serves as a foundation for various aspects of life. Educated individuals tend to have better access to information and are more likely to utilize the health education provided through healthcare institutions. Additionally, women with higher educational

attainment generally enjoy better economic opportunities compared to their uneducated counterparts. Furthermore, we found that female household heads have faced problems of accessing healthcare as compared to male household heads. This might be, female-headed households often face barriers to accessing healthcare, which can be attributed to various factors such as economic, social, educational, healthcare system, psychological, and policy-related issues [32–34]. To bridge this gap, it is necessary to implement comprehensive policies and interventions that specifically address these challenges.

The study found that as women's wealth, and countries income levels increased, they were less likely to have problems accessing healthcare. Previous research had also shown that a person's wealth and financial resources are a key factor in being able to access healthcare. This finding matches what other studies had reported in places like Kenya, Ethiopia, Bangladesh, Myanmar, and across sub-Saharan Africa [13,26,35–37]. The reason for this seems to be that wealthier people can more easily afford to pay for healthcare services. They have the financial means to access the healthcare they need. Another significant finding of this study is that women who had health insurance encountered fewer obstacles when seeking healthcare. This finding supports the healthcare utilization model put forth by Anderson and Newman [38], which posits that having health insurance facilitates easier access to healthcare. Furthermore, it is consistent with earlier research conducted in Ghana [39] and other countries in Sub-Saharan Africa [11], which has shown that possessing health insurance enhances access to maternal healthcare services.

The study also found that a woman's marital status was an important factor in determining the likelihood of facing barriers to accessing healthcare. Compared to women who have never been married, those who are widowed, divorced, or separated are more likely to experience difficulties accessing healthcare. However, married women have lower odds of having problems accessing healthcare. This finding is consistent with previous studies conducted in Bangladesh, Malaysia, Indonesia, and Tanzania [26,40–42]. This may be because married women tend to have better economic and emotional support from their partners, which can help them access healthcare more easily. Being married is also associated with better overall health outcomes, which may be partly due to married women's improved access to and utilization of healthcare services [43].

Women who have more than two children under the age of five and come from larger families were more likely to face challenges in accessing healthcare. This can be attributed to several key factors. Firstly, larger families often have lower monthly incomes, which can make it difficult for women to afford medical care and cover the financial burden of supporting a big household [44,45]. Additionally, women in these situations may have lower awareness about the importance of healthcare services or underestimate the risks associated with not seeking medical attention, leading to delayed or inadequate healthcare utilization [45,46]. Finally, women with multiple young children and extensive caregiving responsibilities often have limited time to prioritize their own medical needs, which can result in delayed or neglected healthcare. Collectively, these financial, awareness, cultural, accessibility, and time constraint factors contribute to the higher odds of women from larger families facing problems in accessing the healthcare they require.

Short birth intervals and adolescent pregnancies can have negative impacts on a woman's ability to access healthcare. This is due to financial limitations, increased physical and emotional demands, maternal health issues, and logistical challenges [47–49]. These factors, combined, make it less likely for women with short birth intervals to access healthcare compared to those with longer intervals who had given first birth after 19 years old. However, by implementing targeted interventions, policies, and support systems, it may be possible to improve healthcare access for women with closely spaced pregnancies.

Women who utilized ANC services and deliver in health facilities generally had better access to healthcare services due to continuity of care, increased health education, access to skilled providers, improved infrastructure, supportive policies, shifting social norms, and financial protection [50–52]. These factors collectively enhance their ability and likelihood to seek and receive healthcare services compared to those who do not follow ANC and deliver at home.

The study revealed that women with greater exposure to mass media, such as radio, newspapers, and television, were less likely to encounter obstacles when accessing healthcare. This finding aligns with previous studies conducted in sub-Saharan Africa [11], Ethiopia [53], Bangladesh [26], and Malawi [54]. This correlation is likely because increased exposure to mass media can enhance a person's health literacy. Health literacy refers to an individual's knowledge, understanding, and skills for making informed decisions regarding their health and healthcare. By improving health literacy, women are empowered to navigate the healthcare system more effectively, comprehend when and how to seek care, and advocate for their health needs. Consequently, this ability helps overcome common barriers women often face, including lack of awareness, cultural factors, and hesitancy to access healthcare.

This study found that women living in rural areas were less likely to be able to access healthcare, compared to women living in urban areas. This matches the findings from previous studies conducted in places like sub–Saharan Africa, Saudi Arabia, the USA, Washington, and Ethiopia [12,36,55,56]. There are a few key reasons why women in rural settings face more difficulties in accessing healthcare: Rural areas often lack good roads, have healthcare facilities located far away, and have limited transportation options. These infrastructural challenges make it harder for rural women to physically get to and use healthcare services. Women in rural communities tend to have lower incomes and less education. This can restrict their ability to afford healthcare costs and understand the importance of seeking medical care. In some cultures, rural women require permission from their husbands before they can seek out healthcare [57]. This added layer of needing approval from a male authority figure can be a major barrier to accessing needed medical services.

## Strength and limitations of the study

The strength of this study lies in its comprehensive approach. By analyzing nationally representative survey data from LMICs, the researchers were able to thoroughly evaluate the challenges women face in accessing healthcare. This aligns with the United Nations' SDGs, providing valuable insights that can inform targeted interventions. A key strength is the study's used of an advanced mixed-effects modeling approach. This allowed the researchers to uncover hidden and previously unexplored factors influencing women's ability to access the healthcare they need. The data was also gathered using standard, verified procedures, adding to the credibility of the findings. As the first study of its kind to focus on the Sustainable Development Goal agenda, this work lays an important foundation. The results can be applied not only to the 61 countries surveyed, but also to other LMICs. The findings will be invaluable for international organizations, regional and national leaders, policymakers, researchers, and healthcare professionals working to improve women's access to care. That said, the study does have some limitations. Its cross-sectional design means causality cannot be established. There may also be biases inherent in the self-reported data from the DHS used. Additionally, certain potentially crucial factors—such as women's own perspectives, sociocultural influences, and issues related to healthcare infrastructure and support programs—were not included in the analysis, due to data constraints. Overall, this comprehensive, rigorous study provides a strong foundation for understanding and addressing the multifaceted barriers to healthcare access

that women in LMICs face. The insights gleaned can guide the development of targeted, context-specific interventions to help achieve the SDGs and improve health outcomes for women globally.

## Conclusions

About two-thirds of women in LMICs have still faced problems to access healthcare. Individual-level factors such as age, education, marital status, wealth, family size, media exposure, birth history, and health insurance coverage play a significant role in determining women's access to healthcare in LMICs. Contextual-level factors, including geographic region, country income level, and literacy rates, also have a substantial impact on women's healthcare access in LMICs. Disparities exist in healthcare access, with certain subgroups of women, such as the poorest, less educated, and those living in rural areas or specific regions, facing greater challenges.

## Recommendations and the way forward

Improving healthcare access for vulnerable women in LMICs will require a multi-faceted approach. We need to develop and implement targeted interventions and policies that address the specific needs of groups like younger women, those with lower education levels, and women from poorer households. For example, we should tailor healthcare service delivery to the unique challenges faced by women in different geographic regions and income levels. This might involve investing in expanding healthcare facilities, especially in rural and underserved areas, to improve physical access. We also need to enhance transportation options and infrastructure to address the mobility challenges women, especially in remote communities, often face. At the same time, expanding and subsidizing health insurance schemes—particularly for the poorest and most vulnerable women—can help reduce financial barriers to accessing care. It's also important that insurance benefits cover a comprehensive range of services, including preventive care and specialized treatments. Enhancing health literacy and education through comprehensive programs can raise awareness and knowledge about available healthcare services, especially among women with lower levels of education. Collaborating with community organizations and local leaders to disseminate health information and promote healthcare-seeking behaviors will be crucial. Addressing social and cultural norms that hinder women's ability to access care is also key. We need to engage with communities to challenge gender-based biases and empower women's decision-making power within households and communities regarding their own healthcare. Strengthening the collection and analysis of sex-disaggregated data on healthcare access and utilization will help us better understand the specific barriers women face. Regularly monitoring and evaluating the effectiveness of interventions will allow us to refine our approaches over time.

Implementing these comprehensive strategies can help ensure more equitable and inclusive access to healthcare for all women in LMICs. This is essential for achieving the Sustainable Development Goals, particularly those related to good health and gender equality. By addressing the individual and contextual barriers, we can improve maternal and child health outcomes and drive progress towards a more just and equitable future.

## Acknowledgments

We would like to acknowledge the DHS program for providing permission for this study following research ethics.

## Author Contributions

**Conceptualization:** Bewuketu Terefe.

**Data curation:** Bewuketu Terefe, Belayneh Shetie Workneh, Alebachew Ferede Zegeye, Tadesse Tarik Tamir, Mulugeta Wassie, Mohammed Seid Ali.

**Formal analysis:** Bewuketu Terefe, Gebreeyesus Abera Zeleke, Mohammed Seid Ali.

**Funding acquisition:** Belayneh Shetie Workneh, Gebreeyesus Abera Zeleke, Enyew Getaneh Mekonen, Alebachew Ferede Zegeye, Agazhe Aemro, Berhan Tekeba, Mulugeta Wassie, Mohammed Seid Ali.

**Investigation:** Gebreeyesus Abera Zeleke, Berhan Tekeba, Mulugeta Wassie.

**Methodology:** Bewuketu Terefe, Belayneh Shetie Workneh, Enyew Getaneh Mekonen, Tadesse Tarik Tamir, Mohammed Seid Ali.

**Project administration:** Gebreeyesus Abera Zeleke, Enyew Getaneh Mekonen, Agazhe Aemro, Mulugeta Wassie, Mohammed Seid Ali.

**Resources:** Belayneh Shetie Workneh, Enyew Getaneh Mekonen, Agazhe Aemro, Tadesse Tarik Tamir, Mulugeta Wassie.

**Software:** Bewuketu Terefe, Tadesse Tarik Tamir.

**Supervision:** Enyew Getaneh Mekonen, Alebachew Ferede Zegeye, Berhan Tekeba, Tadesse Tarik Tamir, Mohammed Seid Ali.

**Validation:** Bewuketu Terefe, Gebreeyesus Abera Zeleke, Alebachew Ferede Zegeye, Agazhe Aemro, Berhan Tekeba, Mohammed Seid Ali.

**Visualization:** Bewuketu Terefe, Belayneh Shetie Workneh, Alebachew Ferede Zegeye, Agazhe Aemro, Berhan Tekeba.

**Writing – original draft:** Bewuketu Terefe, Gebreeyesus Abera Zeleke, Agazhe Aemro, Tadesse Tarik Tamir.

**Writing – review & editing:** Belayneh Shetie Workneh, Gebreeyesus Abera Zeleke, Alebachew Ferede Zegeye, Agazhe Aemro, Berhan Tekeba, Tadesse Tarik Tamir, Mulugeta Wassie, Mohammed Seid Ali.

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
