## [Decision Letter · Decision Letter 0]

7 Oct 2024

PONE-D-24-27452Uncovering Women's healthcare access challenges in low- and middle-income Countries using mixed effects modelling approach: Insights for achieving the Sustainable Development GoalsPLOS ONE

Dear Dr. Terefe,

Thank you for submitting your manuscript to PLOS ONE. After careful consideration, we feel that it has merit but does not fully meet PLOS ONE’s publication criteria as it currently stands. Therefore, we invite you to submit a revised version of the manuscript that addresses the points raised during the review process.

We look forward to receiving your revised manuscript.

Kind regards,

Yibeltal Alemu Bekele, MpH

Academic Editor

PLOS ONE

**Journal Requirements:**

3. Please note that your Data Availability Statement is currently missing the repository name. If your manuscript is accepted for publication, you will be asked to provide these details on a very short timeline. We therefore suggest that you provide this information now, though we will not hold up the peer review process if you are unable.

5. Please upload a new copy of Figure 1 as the detail is not clear. Please follow the link for more information: " ext-link-type="uri" xlink:type="simple">https://blogs.plos.org/plos/2019/06/looking-good-tips-for-creating-your-plos-figures-graphics/"
" ext-link-type="uri" xlink:type="simple">https://blogs.plos.org/plos/2019/06/looking-good-tips-for-creating-your-plos-figures-graphics/"

Reviewers' comments:

Reviewer's Responses to Questions

**Comments to the Author**

1. Is the manuscript technically sound, and do the data support the conclusions?

Reviewer #1: Yes

Reviewer #2: Yes

2. Has the statistical analysis been performed appropriately and rigorously? 

Reviewer #1: Yes

Reviewer #2: Yes

3. Have the authors made all data underlying the findings in their manuscript fully available?

Reviewer #1: Yes

Reviewer #2: Yes

4. Is the manuscript presented in an intelligible fashion and written in standard English?

Reviewer #1: Yes

Reviewer #2: Yes

5. Review Comments to the Author

**Reviewer #1:** The article nicely explores the determining factors for accessibility of health services. I request the authors to clarify on following comments:

1. The study included the data set ranging from 2010 to 2023. As DHS is conducted in every five years, authors need to justify for utilizing the older data like 2011, 2012, 2013, 2014 in different countries.

2. Authors have dependent variables as 'Access to healthcare services' and keep dependent variables as 'ANC visit', 'contraception use' and 'place of delivery' which are the indicators of health service utilization. As accessibility facilitates the environment for utilization of services, authors need to clarify inclusion of the variables of 'health utilization' as independent factors for 'accessibility of health services'.

**Reviewer #2:** - If possible, It's more convenience for reader to read the result when the reference groups are the ideal group according to the author/literature perspective, for example for maternal education to use the secondary higher education as the reference group, as you did in "number of under five" and "family members" variables

- Tables 1 and 2 can be combined

6. PLOS authors have the option to publish the peer review history of their article (what does this mean?). If published, this will include your full peer review and any attached files.

Reviewer #1: No

Reviewer #2: No

---

## [Author Response · Author response to Decision Letter 0]

10 Oct 2024

Response to reviewers 

Dear editor, thank you very much for considering our article in your esteemed journal. Dear reviewers, we would like to thank you for your scholarly contribution to our manuscript. Dear reviewers, we have considered all your comments.

Reviewers' comments:

Reviewer's Responses to Questions

Comments to the Author

1. Is the manuscript technically sound, and do the data support the conclusions?

Reviewer #1: Yes

Reviewer #2: Yes

Response: Dear reviewers, thank you very much for your thorough evaluation of our manuscript. We have responded to your comments below. Thank you again for your valuable feedback.

2. Has the statistical analysis been performed appropriately and rigorously?

Reviewer #1: Yes

Reviewer #2: Yes

Response: Dear reviewers, thank you very much for your thorough evaluation of our manuscript. We have responded to your comments below. Thank you again for your valuable feedback.

3. Have the authors made all data underlying the findings in their manuscript fully available?

Reviewer #1: Yes

Reviewer #2: Yes

Response: Dear reviewers, thank you very much for your thorough evaluation of our manuscript. We have responded to your comments below. Thank you again for your valuable feedback.

4. Is the manuscript presented in an intelligible fashion and written in standard English?

Reviewer #1: Yes

Reviewer #2: Yes

Response: Dear reviewers, thank you very much for your thorough evaluation of our manuscript. We have responded to your comments below. Thank you again for your valuable feedback.

Reviewer 1 

Reviewer: The article nicely explores the determining factors for accessibility of health services. I request the authors to clarify on following comments. The study included the data set ranging from 2010 to 2023. As DHS is conducted in every five years, authors need to justify for utilizing the older data like 2011, 2012, 2013, 2014 in different countries.

Response: Dear reviewer, thank you for your thoughtful feedback on our article. We appreciate your insights regarding the data used in our study. To clarify, we have specifically included the most recent Demographic and Health Surveys (DHS) reports from each country within the time frame of 2010 to 2023. While DHS is typically conducted every five years, some countries may have multiple surveys within this period. In such cases, we have prioritized the latest report available. This approach ensures that our analysis reflects the most current data, thereby minimizing any potential statistical or epidemiological issues related to time span.

Moreover, the rationale for including only the latest DHS reports is rooted in our aim to capture the most accurate and relevant trends in health service accessibility. By focusing on the most recent data, we provide a clearer understanding of current health dynamics and the factors influencing accessibility.

We believe that this method enhances the robustness of our findings and aligns with best practices in epidemiological research. Thank you once again for your valuable input, and we hope this explanation addresses your concerns. Thank you for your insightful comments regarding our variable selection.

Reviewer: Authors have dependent variables as 'Access to healthcare services' and keep dependent variables as 'ANC visit', 'contraception use' and 'place of delivery' which are the indicators of health service utilization. As accessibility facilitates the environment for utilization of services, authors need to clarify inclusion of the variables of 'health utilization' as independent factors for 'accessibility of health services'.

Response: We appreciate your recognition of the importance of distinguishing between accessibility and utilization of healthcare services. In our study, we generated the accessibility variable based on the guidelines from the Guide to DHS Statistics, which serves as a foundational reference for all DHS-based research. This guide informed our outcome variable calculation, focusing on specific questions regarding barriers to accessing healthcare, such as financial constraints, distance to facilities, transport issues, and social factors.

For our outcome variable, we assessed whether at least one of these factors posed a significant challenge for women. If a respondent indicated that any of these issues were a "big problem," we coded their access to healthcare as 1, indicating significant accessibility challenges.

Regarding your point about antenatal care (ANC), postnatal care (PNC), contraception use, and place of delivery, we acknowledge that these are indicators of healthcare utilization rather than direct measures of accessibility. Our intention was to explore how accessibility influences the utilization of these services. While we included these utilization variables to illustrate the broader context of healthcare access, we understand that they should be framed carefully within our analysis.

To clarify, our study primarily focuses on accessibility as a foundational factor that facilitates or hinders healthcare utilization. We aimed to demonstrate how accessibility impacts the likelihood of women engaging with these health services, but we recognize that the utilization variables should not be conflated with access. We appreciate your feedback and ensured this distinction is clearly articulated in our manuscript. Thank you for helping us enhance the clarity of our work.

Reviewer 2

Reviewer : If possible, It's more convenience for reader to read the result when the reference groups are the ideal group according to the author/literature perspective, for example for maternal education to use the secondary higher education as the reference group, as you did in "number of under five" and "family members" variables

Response: Dear reviewer, thank you for your valuable feedback regarding the choice of reference groups in our analysis. We aimed to maintain consistency with existing literature by using the reference groups as established in previous studies. This approach not only aligns our work with established research but also simplifies our discussion points, making it easier for readers to follow our findings in the context of similar studies.

That said, we appreciate your suggestion about using secondary and higher education as the reference group for maternal education. We recognize that this could enhance clarity for the reader and provide a more nuanced understanding of the results. If deemed necessary, we are open to revisiting our choice of reference groups to ensure the most effective presentation of our findings. Thank you for highlighting this aspect, and we will carefully consider it as we finalize our manuscript.

Reviewer: Tables 1 and 2 can be combined

Response: Dear reviewer, we have combined Tables 1 and 2 to enhance clarity and streamline the presentation of our data. This consolidation allows for a more comprehensive view of the information while maintaining readability. Please let us know if you have any further suggestions or concerns regarding this change. Thank you very much

---

## [Decision Letter · Decision Letter 1]

8 Nov 2024

Uncovering Women's healthcare access challenges in low- and middle-income Countries using mixed effects modelling approach: Insights for achieving the Sustainable Development Goals

PONE-D-24-27452R1

Dear Dr. Bewuketu Terefe,

We’re pleased to inform you that your manuscript has been judged scientifically suitable for publication and will be formally accepted for publication once it meets all outstanding technical requirements.

Kind regards,

Yibeltal Alemu Bekele, MpH

Academic Editor

PLOS ONE

Additional Editor Comments (optional):

Reviewers' comments:

Reviewer's Responses to Questions

**Comments to the Author**

1. If the authors have adequately addressed your comments raised in a previous round of review and you feel that this manuscript is now acceptable for publication, you may indicate that here to bypass the “Comments to the Author” section, enter your conflict of interest statement in the “Confidential to Editor” section, and submit your "Accept" recommendation.

Reviewer #1: All comments have been addressed

Reviewer #2: All comments have been addressed

2. Is the manuscript technically sound, and do the data support the conclusions?

Reviewer #1: Yes

Reviewer #2: Yes

3. Has the statistical analysis been performed appropriately and rigorously? 

Reviewer #1: Yes

Reviewer #2: Yes

4. Have the authors made all data underlying the findings in their manuscript fully available?

Reviewer #1: Yes

Reviewer #2: Yes

5. Is the manuscript presented in an intelligible fashion and written in standard English?

Reviewer #1: Yes

Reviewer #2: Yes

6. Review Comments to the Author

Reviewer #1: (No Response)

Reviewer #2: (No Response)

7. PLOS authors have the option to publish the peer review history of their article (what does this mean?). If published, this will include your full peer review and any attached files.

Reviewer #1: **Yes: **Basant Adhikari

Reviewer #2: No

---

## [Editor Report · Acceptance letter]

2 Dec 2024

PONE-D-24-27452R1 

PLOS ONE

Dear Dr. Terefe, 

I'm pleased to inform you that your manuscript has been deemed suitable for publication in PLOS ONE. Congratulations! Your manuscript is now being handed over to our production team.

Kind regards, 

on behalf of

Mr. Yibeltal Alemu Bekele 

Academic Editor

PLOS ONE